# Detection of Pancreatic Cancer miRNA with Biocompatible Nitrogen-Doped Graphene Quantum Dots

**DOI:** 10.3390/ma15165760

**Published:** 2022-08-20

**Authors:** Ryan Ajgaonkar, Bong Lee, Alina Valimukhametova, Steven Nguyen, Roberto Gonzalez-Rodriguez, Jeffery Coffer, Giridhar R. Akkaraju, Anton V. Naumov

**Affiliations:** 1School of Medicine, University of Texas Rio Grande Valley, Edinburg, TX 78539, USA; 2Department of Biology, Texas Christian University, Fort Worth, TX 76129, USA; 3Department of Physics and Astronomy, Texas Christian University, Fort Worth, TX 76129, USA; 4Department of Physics, University of North Texas, Denton, TX 76203, USA; 5Department of Chemistry and Biochemistry, Texas Christian University, Fort Worth, TX 76129, USA

**Keywords:** cancer detection, gene sensing, miRNA, graphene quantum dots, sensor, fluorescence

## Abstract

Early-stage pancreatic cancer remains challenging to detect, leading to a poor five-year patient survival rate. This obstacle necessitates the development of early detection approaches based on novel technologies and materials. In this work, the presence of a specific pancreatic cancer-derived miRNA (pre-miR-132) is detected using the fluorescence properties of biocompatible nitrogen-doped graphene quantum dots (NGQDs) synthesized using a bottom-up approach from a single glucosamine precursor. The sensor platform is comprised of slightly positively charged (1.14 ± 0.36 mV) NGQDs bound via π−π stacking and/or electrostatic interactions to the negatively charged (−22.4 ± 6.00 mV) bait ssDNA; together, they form a complex with a 20 nm average size. The NGQDs’ fluorescence distinguishes specific single-stranded DNA sequences due to bait–target complementarity, discriminating them from random control sequences with sensitivity in the micromolar range. Furthermore, this targetability can also detect the stem and loop portions of pre-miR-132, adding to the practicality of the biosensor. This non-invasive approach allows cancer-specific miRNA detection to facilitate early diagnosis of various forms of cancer.

## 1. Introduction

As the second leading cause of death in the United States, cancer will continue to threaten human health in the coming decades [1]. Even with the advances in cancer therapeutics, timely diagnosis is one of the best methods of averting cancer mortality [2]. Developing effective early cancer detection strategies is critical, as cancer is largely asymptomatic until its later stages. Patient prognoses can be improved with preemptive screening methods, limiting the use of therapeutic cancer treatments and palliative care. Initiatives for early detection of some common types of cancer already exist within the National Breast and Cervical Cancer Early Detection Programs. However, despite pancreatic cancer being the fourth leading cause of cancer-related deaths in the United States, there is neither a widely adopted screening procedure nor a unified non-invasive test for its diagnosis [3]. Historically, over half of pancreatic cancer diagnoses occur at advanced stages, resulting in a low 3% 5-year survival rate [3]. Due to the organ’s location deep inside the body, physical examinations rarely reveal pancreatic tumors, as they often do for other forms of cancer such as breast, cervical, and lung cancers [4,5,6,7]. To effectively detect and diagnose pancreatic cancer, a combination of procedures that include CT scans, MRIs, endoscopic ultrasounds, blood tests, and biopsies is used [4]. Designing a single, accurate, and less invasive testing protocol will enable patients to be diagnosed at earlier stages of pancreatic cancer with better prognoses. Furthermore, segments of the population including the underserved and elderly often lack access to appropriate diagnostic measures, contributing to poorer health outcomes [8,9]. This barrier to effective care highlights the need for early diagnostics that do not involve complex, undesirable, or costly procedures. One promising new avenue can be provided by the detection of cancer-specific microRNA (miRNA).

miRNAs are single-stranded, short (19 to 25 nucleotides) non-coding RNA fragments that play a critical role in regulating gene expression [10,11]. These molecules are transcribed from DNA and later processed into mature miRNAs that bind to closely matching messenger RNA (mRNA) sequences. This interaction marks the mRNA for degradation and represses translation [12]. Overexpressed miRNAs also tend to undergo exocytosis and endocytosis to surrounding cells and dysregulate mRNA function [13]. As many tumors leak miRNAs into the bloodstream, miRNAs can serve as potential biomarkers for the disease [13,14]. Specifically in pancreatic cancer, miRNAs, including miR-34, miR-96, and miR-143, are downregulated, while others, such as miR-132, miR-301a, and miR-454, are upregulated [15,16]. From the list of targetable miRNAs, miR-132 is particularly interesting as studies have shown that its function in pancreatic cancer is to target the tumor suppressor Rb1 mRNA, which promotes cancer tumor growth when inactivated [16]. Furthermore, Khan et al. have demonstrated that, when miR-132 is inhibited, the growth of the tumor slows [15]. Thus, miR-132 appears to be a promising biomarker for detecting pancreatic cancer.

An appropriate diagnostic strategy must be selected to conform to potential targets for detection. miRNAs are commonly detected in biofluids, such as serum, urine, and peritoneal fluid [10]. However, due to the low abundance of miRNAs in these biofluids [17], miRNA detection relies on traditional diagnostic tools, such as polymerase chain reaction (PCR) [18], RNA blotting [19], electrochemistry [20], in situ hybridization [21], and miRNA microarray technologies [22]. Although these assays enable detection in the attomolar and femtomolar ranges, they require additional isolation of miRNAs during purification and amplification steps, leading to false-positive results. These diagnostic technologies are also limited due to the cost of instrumentation and trained personnel required to perform the experiments [17]. Furthermore, routine in-person medical visits needed for screening are hindered due to the low accessibility of hospitals and clinics in remote locations, conflict with work-related schedules, and public health crises, such as the delayed cancer screenings occurring during the SARS-CoV-2 pandemic [23]. To alleviate such diagnostic constraints and ultimately provide efficient, low-cost, and accessible point-of-care cancer detection, we turn to the advances of nanotechnology in biosensing.

Graphene quantum dots (GQDs) have recently attracted attention in the nanotechnology scientific community due to their small average size of 3–5 nm and large surface-to-volume ratio [24,25,26], enabling greater interaction with biological analytes. Unlike their predecessor, graphene, GQDs exhibit photoluminescence (fluorescence) in the visible and, at times, near-infrared, which can be used as a beacon for the detection of biomolecules [27,28]. Fluorescence emission in the visible can originate from GQDs’ size-dependent quantum confinement effect or electronically localized regions on sp^2^ graphitic carbon surrounded by functional groups. Near-infrared emission can arise from smaller confined electronic environments at the functional groups or surface defect states [24,25,29,30]. Partially derivatized graphitic GQD surface provides yet another advantage. While functional groups enable covalent attachment of analytic tools by chemical binding [31,32,33], sp^2^ hybridized regions on the GQDs’ surface enable the loading of aromatic molecules, such as drugs and nucleic acids, via π−π stacking [34,35,36]. This non-covalent route allows the formation of complex nano-biostructures without the need for organic reactions, often involving toxic chemicals [36,37]. Functional groups left intact, on the other hand, render GQDs water-soluble, which is also an important advantage for biomedical applications. Our previous work was focused on developing photostable and highly biocompatible GQDs as prospective imaging agents and therapeutic delivery vehicles [31,38]. However, GQDs’ fluorescence can also be utilized as a sensing tool. High (over 60% [25]) quantum yield emission in the visible with their photostability ensures intense signal from GQD-based sensors and allows for long accumulation times, pushing sensing limits to lower doses. Furthermore, near-infrared emissions capable of penetrating layers of biological tissue may further render GQDs as a basis for implantable sensors. Such nanomaterials, simple and inexpensive in production [25], can become a building block of the sensing technology, addressing the aforementioned critical need in early cancer detection.

In this work, biocompatible and biodegradable nitrogen-doped GQDs (NGQDs) [39] serve as the basis of a simple optical cancer miRNA sensor. These NGQDs are non-covalently bound to bait ssDNA strands that are used as detection probes for miRNA-132 overexpressed in pancreatic cancer. Using short oligonucleotides non-covalently bound to GQDs is advantageous compared to employing larger sequences due to faster and tighter adsorption onto the graphitic platform, as demonstrated with graphene and graphene oxide [40]. The presence of nitrogen dopants in GQDs not only aids with the additional electrostatic complexation with negatively charged oligonucleotides, but also contributes to passivating GQDs’ surface defects that affect their fluorescence [39]. To date, no works utilize the intrinsic GQD or carbon dot fluorescence in miRNA sensing. More often, GQDs and carbon dots are used as intermediary donors for FRET-based fluorescence mechanisms with fluorescent dyes [41,42,43]. However, these dyes may photobleach and are often not suitable for in vivo applications due to their toxicity [30]. Unlike more elaborate synthetic approaches, the NGQD-based detection system used in this work does not use chemical agents other than a single precursor material for NGQD synthesis, which minimizes potential toxicity, cost, and design complexity [37]. The detection system relies solely on non-covalent complexation of NGQDs and ssDNA bait as well as intrinsic NGQD fluorescence to detect pancreatic cancer-generated miRNA-132. Such minimization of structural and functional complexity is expected to yield the desired simplistic low-cost pancreatic cancer sensor architecture. In this work, we design an NGQD-based sensor to assess its feasibility in distinguishing between single-stranded oligos and detecting fragments of miRNA in pancreatic cancer patients versus non-target genes.

## 2. Materials and Methods

### 2.1. Synthesis of Nitrogen-Doped Graphene Quantum Dots (NGQDs)

An amount of 4 g of Glucosamine-HCl (346299, Sigma Aldrich, St. Louis, MO, USA) was mixed with 250 mL of deionized water and treated in the microwave oven (HB-P90D23AP-ST, Hamilton Beach, Southern Pines, NC, USA) for 1 h at a power of 270 W. The suspension, cooled to room temperature, was dialyzed (0.5–1 kDa, Spectrum Chemical Mfg. Corp., New Brunswick, NJ, USA) in deionized water for 2 h with 30 min water changes to remove unreacted precursor material. The product was further freeze-dried (VirTis Freezemobile, 25ES, SP Scientific, Warminster, PA, USA) for storage and further dilution to working concentrations.

### 2.2. Complexation of NGQDs and ssDNA

DNA solution was prepared by mixing 1X TBE (10 mM Tris pH 8, 0.5 M EDTA) buffer with powdered BSP ssDNA (AAG CAA TCA CCA AAA TGA AGA CT, 5′ to 3′) (IDT DNA, Coralville, IA, USA). NGQDs and ssDNA in solution were vortexed for 10 s to facilitate non-covalent complexation without damaging ssDNA through harsher ultrasonic processing. The final concentrations of 1 mg/mL for NGQDs and 27.4 μM for ssDNA were achieved in this process.

### 2.3. Materials Characterization

Fourier Transform Infrared Spectroscopy (FTIR, Thermo Nicolet Nexus, 670, Madison, WI, USA) was used to assess the functional groups present in NGQDs, with measurements performed in the attenuated total reflectance (ATR) mode. The Zeta potential measurement function from the NanoBrook ZetaPALS instrument (Brookhaven Instrument Corporation, NanoBrook, Holtville, NY, USA) was used to evaluate the net charge of NGQDs/ssDNA in an aqueous solution before and after the complexation. Transmission electron microscopy (TEM, JEOL JEM-2100, Peabody, MA, USA) with energy dispersive x-ray analysis (EDS, JEOL, Peabody, MA, USA) was further used to determine NGQDs/ssDNA complexation through their morphology and atomic percentages. Samples were prepared on a carbon-coated 200 mesh copper grid (Electron Microscopy Sciences, CF300-CU, Haltfield, PA, USA) by depositing freeze-dried NGQDs/ssDNA directly onto the grid. Prior to material characterization evaluating the complexation of NGQDs with ssDNA, the sample underwent centrifugal filtration (10 kDa, Southwest Science, HighSpeed™ Microcentrifuge, Roebling, NJ, USA) at 10,000× *g* for 10 min to remove unbound materials. Fluorescence spectral characterization was performed with the Horiba Scientific SPEX NanoLog spectrofluorometer (Nanolog, HORIBA Scientific, Edison, NJ, USA) to assess the oligonucleotide sensing capability of the NGQD/ssDNA complex. Visible fluorescence of the samples was acquired with an excitation wavelength of 400 nm and emission measured in a range of 430 to 650 nm. Subsequent spectral and statistical (*t*-test) analyses were performed in Origin software (OriginLab Corporation, OriginPro 2021b, Northampton, MA, USA).

### 2.4. Fluorescence Test to Assess ssDNA Binding onto the NGQD Platform along with the Hybridization of Complementary, Random, and Pre-miR-132

To facilitate the hybridization of bait ssDNA onto the NGQD platform to its complementary (AGT CTT CAT TTT GGT GAT TGC TT, 5′ to 3′) and random (TCC ATT TGG TTA ATT CGT GTA TC, 5′ to 3′) sequences, NGQDs/ssDNA and the target sequence were annealed at 62.3 °C for 10 min. Similarly, for the hybridization of loop ssDNA132L (TCC AGT TCC CAC, 5′ to 3′) and stem ssDNA132S (CGA CCA TGG CTG TAG ACT GTT A, 5′ to 3′) bait sequences to the complementary loop (C132L, GTG GGA ACT GGA, 5′ to 3′) and stem (C132S, TAA CAG TCT ACA GCC ATG GTC G, 5′ to 3′) ssDNA strands, samples were annealed to 47.2 and 64.8 °C for 10 min, respectively. Lastly, for the hybridization of the pre-miR-132 (miR-132) strand (GAU UGU UAC UGU GGG AAC UGG AGG UAA CAG UC, 5′ to 3′), annealing temperatures of 47.2 and 64.8 °C were used respective to the C132L and C132S bait sequences for 10 min. ssDNA and pre-miR-132 strands were obtained from IDT DNA, Coralville, IA, USA. Sample were annealed using a thermocyler (2720 Thermal Cycler, Applied Biosystems, Waltham, MA, USA).

## 3. Results

In this work, we assembled and tested a pancreatic cancer miRNA sensor based on several components: NGQDs acting as a sensing moiety and bait ssDNA132 that is complementary to the loop part of the pre-miRNA-132 as well as the miRNA duplex region highlighted in pink in Figure 1. The exposed loop on the right is a quasi-single-stranded region where 12 nucleotides (GTG GGA ACT GGA, 5′ to 3′) have the potential for hybridization. Additionally, a double-stranded 22 nucleotide sequence (TAA CAG TCT ACA GCC ATG GTC G, 5′ to 3′) shown in pink can be opened after dehybridization in the cell’s cytoplasm, which becomes available for binding and, thus, detection. To assess the feasibility of this sensing mechanism, we first performed the detection of several ssDNA strands.

To synthesize NGQDs, a bottom-up synthetic approach was used. During the one-step microwave-assisted synthesis, glucosamine molecules undergo a dehydration reaction facilitated by their carboxylic groups. This reaction leads to the polymerization of the precursor and origination of the nucleation centers that ultimately form NGQDs [44]. This simple and affordable synthetic procedure is of interest as it uses biodegradable precursors as the starting material and is free from toxic chemicals typically seen in the complex organic synthesis of nanomaterials. Prior to assembling the pancreatic miRNA sensor, two different bait ssDNAs were used to prove the complexation between ssDNA and the NGQD platform. Indirect assessments of bait NGQDs/ssDNA binding were performed using Zeta potential, transmission electron microscopy (TEM), and high-resolution TEM (HRTEM).

The complexation of NGQDs with ssDNA was performed via co-incubation of these nanostructures, allowing for the adsorption of the nucleotides onto the NGQDs’ graphitic platform, in part via π−π stacking. Post-incubation filtration was used to remove uncomplexed initial materials. This interaction was aided by an electrostatic attraction of the negatively charged phosphate backbone of the bait ssDNA and positively charged amine groups of NGQDs apparent in the FTIR spectra (Appendix A). Zeta potential measurements allowed the assessment of the complexation through the change in net electric charge (Appendix A). With a Zeta potential of 1.14 ± 0.36 mV, NGQDs were incubated with negatively charged bait ssDNA with a Zeta potential of −22.4 ± 6.00 mV. The resulting Zeta potential was −1.48 ± 0.76 mV due to the adsorption of the ssDNA onto the NGQD platform. In addition to the Zeta potential, TEM with HRTEM analysis supported the complexation of NGQDs and ssDNA.

As reported in our previous work [25], NGQDs alone have an average size of 3–5 nm (Figure 2a). Their crystallinity was verified by HRTEM scans (Appendix A), showing an in-plane lattice spacing of 0.22 nm indicative of graphene. Upon complexation with ssDNA, a substantial size increase and some agglomeration was observed (Figure 2b). Moreover, EDS showed the atomic percentages of NGQDs/ssDNA being 82.9% carbon, 4.6% nitrogen, 11.7% oxygen, and 0.73% phosphorus (Figure 2c). The presence of phosphorus, which is not a component of NGQDs, was indicative of ssDNA. At the same time, a low percentage of oxygen indicated that the NGQDs’ surface was not fully covered with functional groups, which allowed for non-covalent binding of ssDNA onto the NGQD graphitic platform.

In this manuscript, the fluorescence of NGQDs was utilized as a sensing mechanism. Thus, to further assess the interaction between NGQDs and ssDNA, the fluorescence emission originating from the NGQDs/ssDNA (AAG CAA TCA CCA AAA TGA AGA CT, 5′ to 3′) sensor was compared to that of the NGQDs alone (Figure 3).

A substantial change in spectral intensity upon co-incubation of NGQDs and ssDNA served as an indicator of their complexation. This complexation is supported by the works of Jeong et al., which demonstrated the successful binding of ssDNA with GQDs of lower oxidation levels, resulting in an altered fluorescence response [36]. The increase in fluorescence intensity upon complexation may result from a charge transfer between ssDNA and the NGQD platform as well as the change in the dielectric environment of NGQDs. Next, the potential of the NGQDs/ssDNA complex for nucleic acid-sensing was analyzed with several tests. The first test assessed the sensor response to the complementary versus random strands, corresponding to dsDNA formation or incomplete binding. It was imperative that a specifically different response be observed with complementary ssDNA strands, as it would indicate the specificity of the sensor and that the device was not merely displaying the result of a higher total ssDNA concentration. The second assessment evaluated the minimum detection capability of the platform. Although the exact concentration of miRNAs in biofluids is not available for pancreatic cancer patients, the experiment aimed to have a high level of sensitivity.

To assess the discrimination between complementary and random ssDNA along with the minimum threshold capability of the NGQDs-based sensor, random and complementary ssDNAs were added to NGQDs/ssDNA sensor. While keeping NGQD concentrations at biocompatible 1 mg/mL [39] and bait ssDNA at 27.4 µM (0.2 mg/mL), the amount of random or complementary ssDNA strands was varied, generating concentration ratios from 1:0.01 up to 1:4 (Figure 4).

Fluorescence of NGQDs after the complexation of those with a bait ssDNA experienced a change in intensity—either an increase or a decrease (Figure 4a,b). This fluorescence change was due to the interaction of NGQDs and bait ssDNA only. Although it was not systematic, as the one arising from the complementary interaction of the bait/target genes, this fluorescence increase/decrease served to indicate the change in the electronic environment of the NGQDs after bait ssDNA attachment. However, further binding of the target strand to the sensor driven by gene complementarity was a more systematic event and caused a monotonic dose-dependent fluorescence response (Figure 4a). Small variations from this trend can arise from more or less energetically favorable gene conformations on the NGQD surface. For example, the 1:0.25 ratio appeared to provide somewhat more efficient complexation than its neighboring concentrations, likely due to the more energetically favorable conformation of the complexed sensor and ssDNA species at this ratio. This trend is not yet a signal of dsDNA formation. However, when a random ssDNA strand was added to the sensor in an aqueous suspension, the dose–response also appeared random (Figure 4b). The initial intensity increase can be contributed to binding more ssDNA strands to the NGQDs, but after the 1:0.1 concentration ratio, the integral fluorescence intensity behavior randomized due to the apparent absence of complementary interaction of the bait sequence on the NGQDs and the one in suspension. When put in this context, the monotonic increase in the sensor fluorescence for the complementary sequence can be interpreted as a result of the hybridization of the bait and target strands on the NGQD surface. Since a detectable difference in intensity occurred for all concentrations tested, a micromolar (2.74 μM) sensitivity can be considered. Due to the scope of this work aiming to discriminate complementary strand form non-specific random gene analyte, one random ssDNA was tested. However, based on the randomized response observed, we expected a similar non-specific effect in the absence of a complementary interaction.

After examining the sensing properties of the platform, tests aimed to assess the detection of specific miRNA were performed. At first, ssDNA132 was now utilized as a bait, complementary to the target pre-miRNA. Pre-miR-132 (Figure 1) is overexpressed in pancreatic cancer patients and has a single-stranded region on its loop alongside a microRNA duplex (stem) that is dehybridized in the cell’s cytoplasm. To test the detection of each part of the target separately, we first tested two target ssDNAs corresponding to the aforementioned miRNA regions. Accordingly, two bait sequences were used, complementary to the target ssDNA representing the loop (ssDNA132L) and stem (ssDNA132S) of pre-miR-132. This test helped validate the ability to hybridize these targets with the sensor prior to testing the full RNA version. A 1:1 concentration ratio of the bait ssDNA132 to the target complementary strands was used as it yielded the greatest NGQDs’ fluorescence difference when targeting the complementary and random sequences (Figure 4).

While a fluorescence increase was observed for both stem and loop targets (Figure 5), indicating a possibility for detection, the responses were minimal and not statistically significant. The fluorescence increase trend in Figure 4a suggests that the sensors were functional for these sequences but not substantially deterministic for utilized concentrations. To assess the sensing potential of the stem and loop on the actual miRNA, NGQD/ssDNA132S and NGQD/ssDNA132L sensors were further subjected to the pre-miRNA-132 (miR-132) at a 1:1 ratio.

Interestingly, a greater increase in fluorescence intensity was observed for both loop and stem sensors, detecting the actual miR-132 rather than its smaller DNA counterparts. This difference is potentially attributed to the charge transfer between NGQDs and a larger ribonucleic acid, which enables deterministic detection of both the loop and stem of the target miRNA at a concentration of 27.4 µM. The following results suggest that miR-132 can be detected using the proposed NGQD platform, which is the objective of this work. To define the stability of the sensor, the fluorescence of NGQDs/ssDNA132S without and with the complexation with miR-132 were measured within 25 h (Appendix A). A similar decrease in intensities was observed within the first 3 h for both NGQDs/ssDNA132S and NGQDs/ssDNA132S + miR-132. However, within the next hours, their fluorescence intensities remained stable. This trend indicates that miR-132 is not detached from NGQDs/ssDNA132 to alter the fluorescence of NGQDs/ssDNA132S + miR-132, and the sensor continues to successfully detect the presence of the analyte. Furthermore, to determine the sensitivity of the NGQDs/ssDNA132 sensor in detecting miR-132, the stem and loop sensors were treated with a range of miR-132 concentrations down to 0.274 µM for a 1:0.01 ratio between the bait ssDNA132 and miR-132 (Figure 6).

In the chosen range, both sensors generally showed a monotonic signal increase with greater analyte concentrations. Intensity fluctuations at low concentrations between the 1:0.01 and 1:0.1 ratios for the loop sensor were generally within the error bars and can occur if lower ratios provide slightly more energetically favorable configuration. The biosensor targeting the stem of miR-132 had an initial statistically significant response (*p* < 0.05) at the 1:0.1 concentration ratio, corresponding to the 2.74 µM miR-132 concentration. On the other hand, for the biosensor targeting the loop of miR-132, a statistically significant response (*p* < 0.05) was initially observed at 1:0.25, corresponding to the 6.85 µM miR-132 concentration. These results suggest the initial and progressive detection of the analyte. At the same time, the minimal sensitivity of the NGQD/ssDNA132 sensor remains in the micromolar range, which is suitable for diagnosing the onset and progression of cancer from ex vivo samples, such as cells and tissues [18,45,46,47]. However, most importantly, this work verifies the capability of using GQDs’ intrinsic properties for miRNA detection and sets the benchmark to advance the detection limit towards the attomolar range, which is suitable for sensing in biofluids [48].

## 4. Conclusions

In this work, we developed a low-cost biocompatible and nitrogen-doped graphene quantum dots (NGQDs) sensor for pancreatic cancer miRNA (miRNA-132). The sensing mechanism is based on the variation of NGQDs’ intrinsic fluorescence intensity with the complementary binding of the target gene to the bait ssDNA on the NGQD surface. NGQDs are synthesized via a simple one-step hydrothermal microwave-assisted process from a single biocompatible glucosamine precursor, which makes them water-soluble and biocompatible at concentrations over 1 mg/mL. Zeta potential measurements suggest that the detection platform, NGQD, is positively charged (1.14 ± 0.36 mV) and the bait ssDNA sensing sequence is negatively charged (−22.4 ± 6.00 mV), where ssDNA binds electrostatically and via π − π stacking onto the NGQD’s graphitic surface. Binding with the bait ssDNA sequence altered the fluorescence intensity of the NGQD, likely due to charge transfer from ssDNA as it formed 20 nm complexes. The specificity and sensitivity of the resulting NGQDs/ssDNA sensor were tested against ssDNA and miRNA targets. The sensor showed a specific ssDNA dose-dependent response only to the complementary target, with minimal sensitivity in the micromolar range. While the platform also showed some response to the ssDNA sequences mimicking the loop and stem single-stranded parts of the miRNA, its response to the actual miRNA was more dramatic. Both sensors for the stem and loop parts of the miRNA-132 exhibited dose-dependence at the corresponding minimal concentrations of 2.74 and 6.85 µM. This detection capability warrants the potential for ex vivo cancer miRNA detection with the advantages of being low-cost, simple, and non-invasive. This developed biosensor addresses multiple critical needs of cancer diagnostics and sets the groundwork for further development of attomolar gene concentration sensing in vivo.

## Figures and Tables

**Figure 1 materials-15-05760-f001:**
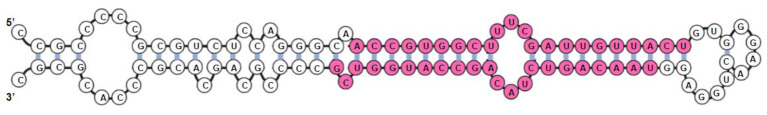
The nucleotide structure of pre-miR-132 with quasi-single-stranded loop (on the right) and pink-highlighted stem duplex region considered as potential binding targets.

**Figure 2 materials-15-05760-f002:**
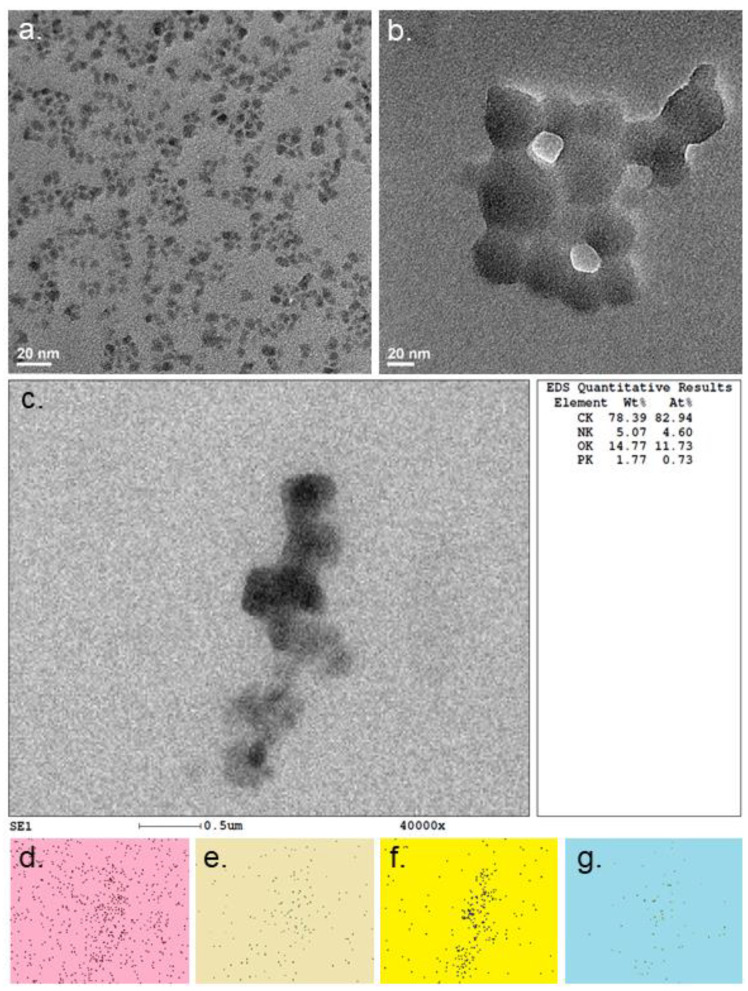
TEM of (**a**) NGQDs and (**b**) NGQDs/ssDNA. EDS maps of (**d**) carbon, (**e**) nitrogen, (**f**) oxygen, and (**g**) phosphorous in the area of the TEM scan in (**c**), and the table of atomic percentages of carbon, nitrogen, oxygen, and phosphorous in that region.

**Figure 3 materials-15-05760-f003:**
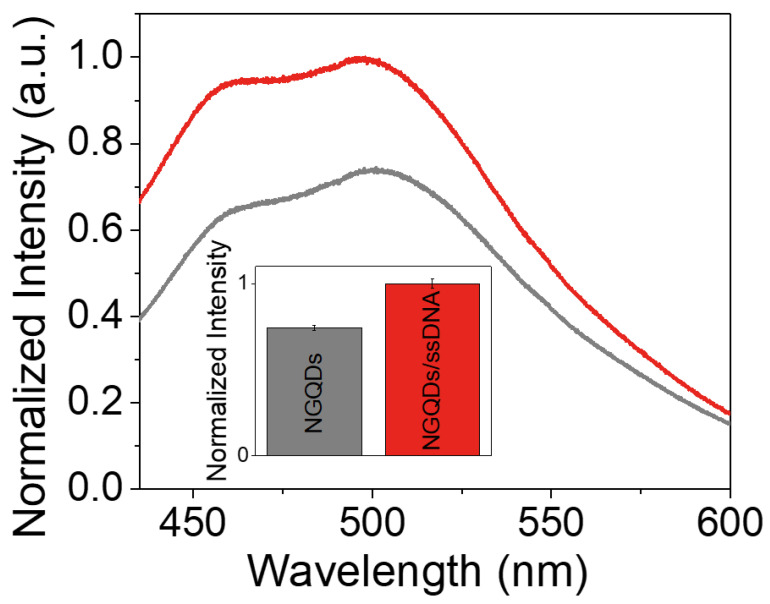
Spectral changes upon interaction between NGQDs and ssDNA. Fluorescence spectra of NGQDs (grey) at 1 mg/mL and NGQDs mixed with ssDNA (NGQDs/ssDNA) (red) at 1 mg/mL: 27.4 µM along with their integrated intensities shown as a bar plot.

**Figure 4 materials-15-05760-f004:**
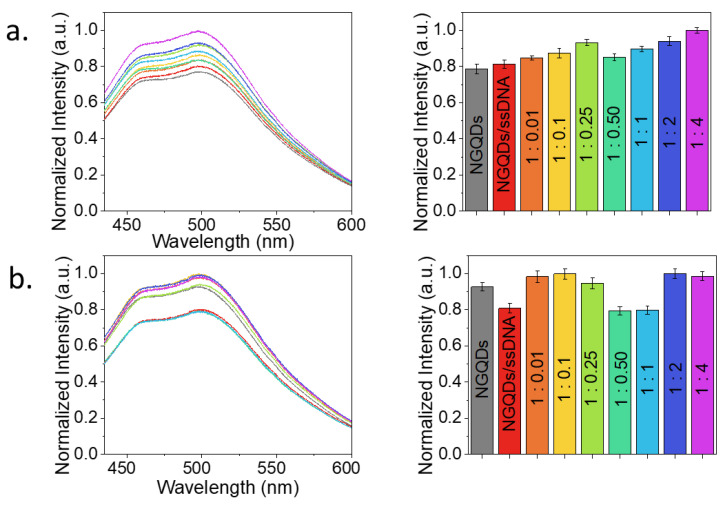
Detection of complementary (**a**) and random (**b**) ssDNA utilizing NGQDs/ssDNA platform. The fluorescence spectra of NGQDs, NGQDs/ssDNA, and NGQDs/ssDNA mixed with complementary and random ssDNA strands at the concentration ratios of 1:0.01 (orange), 1:0.1 (yellow), 1:0.25 (green), 1:0.5 (teal), 1:1 (light blue), 1:2 (blue), and 1:4 (purple) are shown along with their corresponding integrated intensities displayed as bar plots.

**Figure 5 materials-15-05760-f005:**
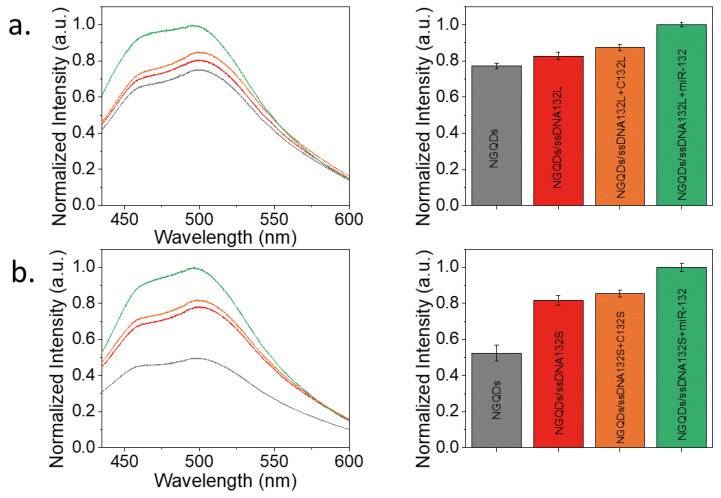
The detection of loop (**a**) and stem (**b**) of pre-miR-132 and its ssDNA version. Fluorescence spectra of NGQDs (grey), NGQDs complexed with the bait for either the loop or stem ssDNA132L/S (red), NGQDs complexed with the loop or stem bait (ssDNA132L/S) mixed with their respective complementary ssDNA strands (C132L/S) (orange) or miR-132 (green). Corresponding integrated spectral intensities are shown as bar plots.

**Figure 6 materials-15-05760-f006:**
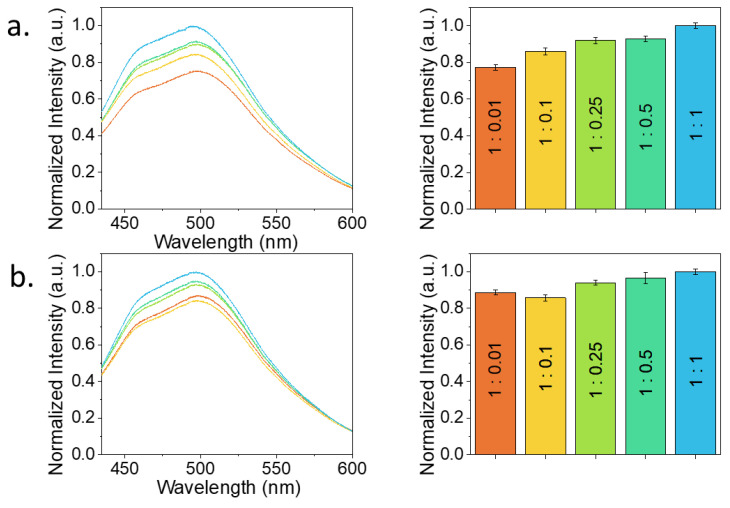
Threshold fluorescence sensitivity assessment of NGQDs/ssDNA132 sensor for miR-132. Fluorescence of (**a**) NGQDs/ssDNA132S and (**b**) NGQDs/ssDNA132L mixed with miR-132 at the concentration ratios of 1:0.01 (orange), 1:0.1 (yellow), 1:0.25 (green), 1:0.5 (teal), 1:1 (light blue). Corresponding integrated spectral intensities shown as bar plots.

## Data Availability

The data presented in this study are available on request from the corresponding author. The data are not publicly available due to privacy.

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
