# Peer review of "Detection of Pancreatic Cancer miRNA with Biocompatible Nitrogen-Doped Graphene Quantum Dots"

_materials, 2022, doi:10.3390/ma15165760_

Round 1
Reviewer 1 Report
Authors report on the “Detection of Pancreatic Cancer miRNA with Biocompatible Nitrogen-Doped Graphene Quantum Dots”. In this work, the presence of specific pancreatic cancer-derived miRNA (pre-miR-132) is detected using the fluorescence properties of biocompatible nitrogen-doped graphene quantum dots (NGQDs) synthesized using a bottom-up approach from a single glucosamine precursor. The sensor platform is comprised of slightly positively charged (1.14±0.36 mV) NGQDs bound via ?−? stacking and/or electrostatically to the negatively charged (- 22.4±6.00 mV) bait ssDNA; together forming a 20 nm average-sized complex. NGQDs’ fluorescence distinguishes specific single-stranded DNA sequences due to their bait-target complementarity, discriminating them from random control sequences with sensitivity in the micromolar range. Furthermore, this targetability can also detect the stem and loop portions of pre-miR-132, adding to the practicality of the biosensor. The work is interesting but needs a bit revision before it will be suitable for publication.
1. There may be a mistake in Figure 6b. What does the “1:01” means?
2. According to the result of Figure 6b, the fluorescence value at the concentration ratio of 1:0.01 is higher than that at the concentration ratio of 1:0.1. Please explain it.
3. What is the basis for choosing the concentration ratios? In other words, why are the concentration ratios 1:0.01, 1:0.1, 1:0.25, 1:0.5 and 1:1?
4. Stability is a very important indicator of the sensor. It is suggested to supplement the stability experiment.
Author Response
Response to Reviewer 1 Comments
Reviewer 1:
Authors report on the “Detection of Pancreatic Cancer miRNA with Biocompatible Nitrogen-Doped Graphene Quantum Dots”. In this work, the presence of specific pancreatic cancer-derived miRNA (pre-miR-132) is detected using the fluorescence properties of biocompatible nitrogen-doped graphene quantum dots (NGQDs) synthesized using a bottom-up approach from a single glucosamine precursor. The sensor platform is comprised of slightly positively charged (1.14±0.36 mV) NGQDs bound via ?−? stacking and/or electrostatically to the negatively charged (- 22.4±6.00 mV) bait ssDNA; together forming a 20 nm average-sized complex. NGQDs’ fluorescence distinguishes specific single-stranded DNA sequences due to their bait-target complementarity, discriminating them from random control sequences with sensitivity in the micromolar range. Furthermore, this targetability can also detect the stem and loop portions of pre-miR-132, adding to the practicality of the biosensor. The work is interesting but needs a bit revision before it will be suitable for publication.
Point 1: There may be a mistake in Figure 6b. What does the “1:01” means?
Response 1: We thank the reviewer for pointing out this typo. 1:01 was intended as 1:0.01. We have corrected it and added the corrected bar plot to Figure 6b in the revised version.
Point 2: According to the result of Figure 6b, the fluorescence value at the concentration ratio of 1:0.01 is higher than that at the concentration ratio of 1:0.1. Please explain it.
Response 2: We thank the reviewer for pointing out the need for clarification and add the following discussion on page 11 of the manuscript:
Intensity fluctuations at low concentrations between 1:0.01 and 1:0.1 ratios for the loop sensor are generally within the error bars and can occur if lower ratios provide slightly more energetically favorable configuration.
Point 3: What is the basis for choosing the concentration ratios? In other words, why are the concentration ratios 1:0.01, 1:0.1, 1:0.25, 1:0.5 and 1:1?
Response 3: We would like to thank the reviewer for pointing this out. The concentration ratios were chosen to span µM concentrations, suitable for diagnosing the onset and progression of cancer from ex vivo samples [18, 45-47].
Point 4: Stability is a very important indicator of the sensor. It is suggested to supplement the stability experiment.
Response 4: Upon the reviewer’s suggestion, the stability experiment has been performed and the data is added to the Supplementary Material (Figure S4). The results of this stability experiment are reported in the main manuscript (page 10) as follows:
To define the stability of the sensor, the fluorescence of NGQDs/ssDNA132S without and with the complexation with miR-132 were measured within 25 hours (Figure S4). Similar decrease in intensities was observed within the first 3 hours for both NGQDs/ssDNA132S and NGQDs/ssDNA132S+miR-132. However, within the next hours, their fluorescence intensities remain stable. This trend indicates that miR-132 is not detached from NGQDs/ssDNA132 to alter the fluorescence of NGQDs/ssDNA132S+miR-132 and the sensor continues to successfully detect the presence of the analyte.

Reviewer 2 Report
In this manuscript, the authors reported a bottom-up approach from a single glucosamine precursor to synthesize biocompatible nitrogen-doped graphene quantum dots (NGQDs) with excellent fluorescence properties and its sensors for pancreatic cancer miRNA (miRNA-132) has been constructed. In this work, the authors have conducted detailed analysis but some minor mistakes need to be corrected before acceptance.
Author Response
Response to Reviewer 2 Comments
Reviewer 2:
In this manuscript, the authors reported a bottom-up approach from a single glucosamine precursor to synthesize biocompatible nitrogen-doped graphene quantum dots (NGQDs) with excellent fluorescence properties and its sensors for pancreatic cancer miRNA (miRNA-132) has been constructed. In this work, the authors have conducted detailed analysis but some minor mistakes need to be corrected before acceptance.
Response: We thank the reviewer for their comment and advice. We have introduced extensive edits to the manuscript in order to address all minor mistakes.
Reviewer 3 Report
This study explores the development of an optical microRNA sensor based on fluorescence of graphene quantum dots (GQDs). The sensor uses immobilized DNA that is non-covalently conjugated to the GQDs through pi-stacking and hybridized with complementary DNA and microRNA strand, resulting in fluorescence changes. They apply this sensor to detect microRNAs associated with pancreatic cancer.
The manuscript is written clearly and easy to follow. The research approach is novel and the topic has great potential for impact in the community. Though the findings show mixed results, the authors are forthcoming in the limitations of their data, which I believe is worthy of publishing. It would help for the authors to tone down some of the claims in the manuscript to reflect more accurately some of the results. I therefore recommend publishing this paper with minor revisions summarized below.
1. In Figure 3, authors claim that the PL intensity of GQDs is less than that of their corresponding complexes with ssDNA when they have same concentrations (1mg/ml). How do they make sure that the final concentration of GQD in GQD/DNA complex is 1mg/ml? Because after complexation, the 10 kD filter is used to remove unbound GQDs, which could remove some of the material. Even if the sample is freeze-dried, they should specify how they verify the GQD contribution to the weight of the total complex.
- Is there any way to know how many DNA strands are adsorbed onto a single GQD surface? Previous reports by Jeong and Landry et al. indicate that the DNA does not spontaneously conjugate with the GQD under certain conditions (such as high oxidation), though the authors appear to cite this manuscript to indicate the opposite. This should be clarified in the text.
- Why does the PL intensity decrease from 1:0.1 to 1:0.25 (or 1:0.25 to 0.5) in Fig4a? Is there a change in the mechanism?
- Why is the PL intensity of the complex less than the GQD itself in Fig4b? Is a different ssDNA (than panel a) used in the complexation step?
- Line 325: “pre-miR-132” is written twice.
- How do the authors ensure that ssDNA is not detached from GQDs after hybridization? Have they checked experimentally if the complex is stable after adding the complementary strands?
- The authors should include a brief discussion on the randomness of the response they observed for the random DNA sequences for different random sequences (How consistent are the observed trends for the random sequences? Are different random sequences expected to follow the same trend or are they expected to give different random results?)
- The authors specify in the conclusions that the sensor shows “substantial specificity”, when their results confirm that they also get responses in the negative control with the random sequence. They should therefore consider revising this claim, particularly since the random microRNA sequence was not tested in Figure 5 as was done with the ssDNA.
Author Response
Response to Reviewer 3 Comments
Reviewer 3:
This study explores the development of an optical microRNA sensor based on fluorescence of graphene quantum dots (GQDs). The sensor uses immobilized DNA that is non-covalently conjugated to the GQDs through pi-stacking and hybridized with complementary DNA and microRNA strand, resulting in fluorescence changes. They apply this sensor to detect microRNAs associated with pancreatic cancer.
The manuscript is written clearly and easy to follow. The research approach is novel and the topic has great potential for impact in the community. Though the findings show mixed results, the authors are forthcoming in the limitations of their data, which I believe is worthy of publishing. It would help for the authors to tone down some of the claims in the manuscript to reflect more accurately some of the results. I therefore recommend publishing this paper with minor revisions summarized below.
Point 1: In Figure 3, authors claim that the PL intensity of GQDs is less than that of their corresponding complexes with ssDNA when they have same concentrations (1mg/ml). How do they make sure that the final concentration of GQD in GQD/DNA complex is 1mg/ml? Because after complexation, the 10 kD filter is used to remove unbound GQDs, which could remove some of the material. Even if the sample is freeze-dried, they should specify how they verify the GQD contribution to the weight of the total complex.
Response 1: We thank the reviewer for pointing this out. Here, we would like to clarify that while NGQDs were purified after their synthesis, NGQDs/ssDNA sensor complex was filtered only prior to Zeta potential and TEM/EDS measurements and not prior to PL studies. In the case of PL sensing measurements, the concentration of 1 mg/ml is based on the amount of material that was put in. In order to address confusing language in our manuscript, we have moved the information about sample filtration to the Materials Characterization section (page 4):
Prior to Zeta potential and TEM/EDS measurements evaluating the complexation of NGQDs with ssDNA, the sample undergoes centrifugal filtration (10 kDa, Southwest Science, HighSpeed™ Microcentrifuge, Roebling, NJ, USA) at 10,000xg for 10 min to remove unbound materials.
Point 2: Is there any way to know how many DNA strands are adsorbed onto a single GQD surface? Previous reports by Jeong and Landry et al. indicate that the DNA does not spontaneously conjugate with the GQD under certain conditions (such as high oxidation), though the authors appear to cite this manuscript to indicate the opposite. This should be clarified in the text.
Response 2: We would like to thank the reviewer for this important comment. Although it is unclear how much ssDNA is absorbed onto a single NGQD surface, we do observe the ssDNA attachment by the change in NGQDs’ fluorescence. This indicates the adsorption of the ssDNA on the GQD surface. According to the EDS results (Figure 2), the amount of oxygen content in NGQDs doesn’t exceed 12 at. % indicating lower oxidation limits. Thus, our results are not expected to contradict the study by Jeong et al. Upon reviewer’s suggestion, we have introduced corrections into the statement on page 7 of the main manuscript to outline that complexation happens at lower GQD oxidation levels:
This complexation is supported by the works of Jeong et al. that have demonstrated successful binding of ssDNA with GQDs of lower oxidation levels, resulting in an altered fluorescence response [36].
Point 3: Why does the PL intensity decrease from 1:0.1 to 1:0.25 (or 1:0.25 to 0.5) in Fig4a? Is there a change in the mechanism?
Response 3: We follow the reviewer’s comment, and add the following discussion on page 8 of the manuscript to explain this:
Small variations from this trend can arise from more or less energetically favorable conformations of the genes on the NGQD surface. For example, 1:0.25 ratio appears to provide somewhat more efficient complexation than its neighboring concentrations, likely due to the more energetically favorable conformation of the complexed sensor and ssDNA species at this ratio.
Point 4: Why is the PL intensity of the complex less than the GQD itself in Fig4b? Is a different ssDNA (than panel a) used in the complexation step?
Response 4: We thank the reviewer for pointing out this deficiency in our explanation. Upon the reviewer’s comment, we have added the following discussion on page 8 to address it:
Fluorescence of NGQDs after the complexation of those with a bait ssDNA experiences a change in intensity, either increase or decrease (Figures 4a and 4b). This fluorescence change is due to the interaction of NGQDs and bait ssDNA only. Although it is not systematic, as the one arising from complementary interaction of the bait/target genes, this fluorescence increase/decrease serves to indicate the change in the electronic environment of the NGQDs after bait ssDNA attachment. However, further binding of the target strand to the sensor driven by gene complementarity is a more systematic event and causes a monotonic dose-dependent fluorescence response (Figure 4a).
Point 5: Line 325: “pre-miR-132” is written twice.
Response 5: Upon reviewer’s suggestion this statement is corrected:
Figure 5. The detection of loop (a) and stem (b) of pre-miR-132 and its ssDNA version.
Point 6: How do the authors ensure that ssDNA is not detached from GQDs after hybridization? Have they checked experimentally if the complex is stable after adding the complementary strands?
Response 6: Upon the reviewer’s suggestion, the stability experiment has been performed and the data is added to the Supplementary Material (Figure S4). The results of this stability experiment are reported in the main manuscript (page 10) as follows:
To define the stability of the sensor, the fluorescence of NGQDs/ssDNA132S without and with the complexation with miR-132 were measured within 25 hours (Figure S4). Similar decrease in intensities was observed within the first 3 hours for both NGQDs/ssDNA132S and NGQDs/ssDNA132S+miR-132. However, within the next hours, their fluorescence intensities remain stable. This trend indicates that miR-132 is not detached from NGQDs/ssDNA132 to alter the fluorescence of NGQDs/ssDNA132S+miR-132 and the sensor continues to successfully detect the presence of the analyte.
Point 7: The authors should include a brief discussion on the randomness of the response they observed for the random DNA sequences for different random sequences (How consistent are the observed trends for the random sequences? Are different random sequences expected to follow the same trend or are they expected to give different random results?)
Response 7: We thank the reviewer for this valuable suggestion. The goal of this particular work was to discriminate complementary strands from some non-particular random strand, thus, one random strand was used. We have added a discussion to page 9 of the main manuscript to clarify this:
Due to the scope of this work aiming to discriminate complementary strand form non-specific random gene analyte, one random ssDNA is tested. However, based on the randomized response observed, we expect similar non-specific effect in the absence of complementary interaction.
Point 8: The authors specify in the conclusions that the sensor shows “substantial specificity”, when their results confirm that they also get responses in the negative control with the random sequence. They should therefore consider revising this claim, particularly since the random microRNA sequence was not tested in Figure 5 as was done with the ssDNA
Response 8: We would like to thank the reviewer for pointing this out, we agree that it might be confusing for future readers. While we are not testing random microRNA, we do observe concentration dependence for the case of complementary ssDNA and not for the random one. We have revised and modified the conclusion on page 11 to address reviewer’s comment:
Sensor shows a specific ssDNA dose-dependent response only to the complementary target with minimal sensitivity in the micromolar range.